# Anti-neuroinflammatory and neuroprotective potential of *Cissus tuberosa* ethanol extract in Parkinson's disease model through the modulation of neuroinflammatory markers

Malik Saadullah[1], Amna Sehar[1], Zunera Chauhdary🄳[2]*, Rida Siddique[2], Hafsa Tariq[1], Muhammad Asif[3], Shazia Anwer Bukhari[4], Aisha Sethi[5]

1 Department of Pharmaceutical Chemistry, Government College University Faisalabad, Faisalabad, Pakistan, 2 Department of Pharmacology, Government College University Faisalabad, Faisalabad, Pakistan, 3 Department of Pharmacology, Islamia University Bahawalpur, Bahawalpur, Pakistan, 4 Department of Biochemistry, Government College University Faisalabad, Faisalabad, Pakistan, 5 Department of Pharmaceutics, Government College University Faisalabad, Faisalabad, Pakistan

* zunerach@yahoo.com

## Abstract

The plant *Cissus tuberosa Moc* is abundant in phenolics, has been documented to have neuroprotective properties. The study seeks to determine the neuroprotective effects of *C. tuberosa* ethanolic extract (CTE) against Parkinson's disease by evaluating its impact on motor dysfunction, cognitive deficits, neuroinflammation, and neurodegeneration in paraquat-induced Parkinson's disease models. The research hypothesizes that CTE can modulate key biomarkers involved in Parkinson's pathology, including α-synuclein, interleukin-1β (IL-1β), and tumor necrosis factor-α (TNF-α), assessed through qRT-PCR, as well as interleukin-6 (IL-6) and TNF-α, evaluated through ELISA. Parkinson disease was induced by using paraquat intraperitoneally. The study was designed by considering various groups with their respective treatments, control group treated normally, disease control receiving paraquat (1 mg/kg, i.p.), standard treated grabbed with (levodopa+carbidopa), and three treatment groups received plant extract (150, 300, 600 mg/kg) respectively for 21 days study period. Both behavioral, and biochemical analysis were performed. HPLC analysis revealed the presence of several phenolic compounds. CTE significantly improved motor function and cognitive performance in rats, showing a dose-dependent reduction in paraquat-induced neurotoxicity (150 < 300 < 600 mg/kg, P<0.001). CTE significantly restored antioxidant enzyme levels (P<0.001), contributing to the alleviation of oxidative stress. Neurotransmitter levels were significantly improved in a dose-dependent manner (P<0.001), while acetylcholinesterase (AChE) levels were significantly reduced (P<0.001). CTE treatment showed significant restoration of brain tissue, reducing neuroinflammation and neurodegeneration, thereby preserving normal brain structure. ELISA testing demonstrated a significant (P<0.001) downregulation of IL-6 and TNF-α levels in CTE-treated groups. qRT-PCR results showed significant downregulation of α-synuclein, IL-1β, and TNF-α mRNA expression in CTE-treated groups compared to the diseased group, suggesting neuroprotective effects. The study concludes that CTE has potential therapeutic effects in alleviating

**Data Availability Statement:** All relevant data are within the paper.

**Funding:** The author(s) received no specific funding for this work.

**Competing interests:** The authors have declared that no competing interests exist.

**Abbreviations:** AChE, Acetyl Cholinesterase; AD, alveolar duct; BD, bile duct; BR, bronchiole; CAT, catalase; CN, central nuclei; CTE, *Cissus tuberosa* ethanolic extract; CV, central vein; DCT, distal convoluted tubules; HA, hepatic artery; HPLC, High-performance liquid chromatography; IL, interleukin; MF, myocardial fibrils; PCT, proximal convoluted tubules; PD, Parkinson's disease; PV, portal vein; RP, red pulp MDA, malondialdehyde; SNpc, substantia nigra pars compacta; SOD, Superoxide dismutase; TBA, Thiobarbituric acid; TCA, trichloroacetic acid; TNF- α, tumor necrosis factor; US, urinary space; WP, white pulp.

Parkinson's disease symptoms, primarily through its antioxidant, anti-inflammatory, and neuroprotective properties.

## Introduction

Parkinson's disease (PD) is a neurodegenerative disorder that typically manifests with age, leads to neuron deterioration, substantia nigra neuron loss, protein aggregates, and neuro-inflammation. This results in motor dysfunction due to reduced dopamine in specific brain regions [1]. Persistent neuro-inflammation, driven by oxidative stress as well as activated glial cells [2, 3], are closely associated with cell death in Parkinson's disease, potentially leading to gradual asymptomatic stage impairment [4]. Epidemiological and genetic research confirms neuroinflammation's role in PD development [5]. Neuro-inflammation may be triggered by environmental toxins such as paraquat [6].

Neuroinflammation takes place when microglia and reactive astrocytes within the central nervous system (CNS) become stimulated, leading to the release of inflammatory agents such as cytokines, chemokines, prostaglandins, complement proteins, and reactive oxygen/nitrogen compounds. This activation has the capacity to undermine the integrity of the blood-brain barrier [7]. Over the progression of the illness, resident brain macrophages go through a change, becoming activated antigen-presenting cells [8], displayed over expression of proin-flammatory mediators like TNF-α, IL-1, and IL-6, found in the cerebrospinal fluid (CSF) and brain parenchyma of PD development. Significantly, there is a notable increase in microglial proliferation, This is particularly evident in the striatum and substantia nigra (SN), underscor-ing their pivotal role in the neuroinflammatory mechanisms associated with PD [9].

Biomarkers play an essential role in the early detection, monitoring, and assessment of PD. Molecular indicators, such as α-synuclein found in cerebrospinal fluid (CSF) and other bodily fluids, are being investigated. Additionally, potential biomarkers for neuroinflammation in PD may be associated with inflammatory molecules. CSF is a valuable source for uncovering molecular changes linked to neurodegeneration, while peripheral blood may reveal inflamma-tory factors from affected brain regions, providing insights into PD's role in its development. The aggregation of the aberrant, insoluble form of α-synuclein significantly influences PD pro-gression, This may entail the presence of misfolded α-synuclein within pathogen-associated molecular patterns or damage-associated molecular patterns [10]. Increased proinflammatory cytokine level, interleukin-1β in the striatum are linked to the death of dopaminergic neurons and the development of motor impairments when this elevation persists [11]. In PD progres-sion, increased IL-1β levels are observed in the striatum, cerebrospinal fluid, and serum, par-ticularly in individuals with sleeping disorders [12] and high antibody titers against common pathogens [13]. Interleukin-6, a versatile cytokine primarily produced by neurons and glial cells, is linked with more risks of PD, when there are elevated plasma levels of this pro-inflam-matory cytokine [14]. In individuals with PD, there is a negative relationship between serum IL-6 levels and clinical measures such as functional mobility, gait speed, and scores on the mini-mental status examination [15].

Tumor necrosis factor alpha (TNF-α), a pro-inflammatory cytokine crucial for the body's immune response, initiates the activation of microglia, resulting in the gradual loss of dopami-nergic neurons in the SN. Individuals with PD demonstrate an increased expression of TNF-α in both the SN and cerebrospinal fluid [16, 17]. Acetylcholinesterase (AChE) primarily serves to conclude cholinergic neurotransmission at synapses through the degradation of

acetylcholine. An increase in the expression of a distinct synaptic form of AChE mRNA is associated with the initiation of apoptosis in different cell lines, and the levels of both AChE and its mRNA rise in reaction to neurotoxic influences [18].

Plant originated phytochemical plays pivotal character to maintain brain's chemical balance to influence the receptor functions for main inhibitory neurotransmitters [19]. Recent researches revealed that the consumption of phyto nutritional elements have a significant impact on the agingbrain, perhaps improving cognition and motor skills [20]. A plant genus *Cissus*, species are widely distributed in Asia, traditionally, it possess strong potential to treat skin disorders, bone issues, cardiovascular ailments, pyrexia, digestive ailments, epilepsy, respiratory problems and eye diseases [21]. Unique medicinal potentials linked to the genus *Cissus* due to their active secondary metabolites, including flavonoides, alkaloids, lipids, fatty acids, terpenes, iridoides, saponins, steroids, chalcones, glycosides, vitamins and acids. It is verified from extensive literature survey the medicinal potential of *Cissus tuberosa* least reported against neuroprotection. This study was established to explore the untapped therapeutic potential through chemical characterization. Effectiveness for anti-parkinsonism was observed against paraquat induced rat model by ELISA and qRT-PCR.

## Material and methods

### Plant collection and extraction

In September 2022, *C.tuberosa* collected from (Faisalabad) nursery. The collected specimen was authenticated by taxonomist given a herbarium number: 424/GCUF, Government College University Faisalabad, Pakistan. Maceration technique was employed for extraction. The plant material was cleaned to remove dirt, and seprated leaves were dried at room temperature under shade for 15 days. The dried material was subjected to grinder to make fine powder. By using ratio (3:1), 1500 mL of ethanol 500 g dry plant powder was suspended, left to stand with light shaking. After 24 hours the material was filtered by filter paper. Macerated material was filtered three times to obtain maximum quantity of extract. After 72 hours, menstruum was placed in rotary at standard condition (temperature < 40 ˚C, pressure < 70 torr.) for evaporation. Dried extract was placed in a clean air tight glass container. Percentage yield determined by given formula [22].

$$\text{Percentage yield} = \frac{\text{Dry weight of the extract}}{\text{Dry weight of the plant material}} \times 100$$

### Phytochemical screening

The phytochemical screening of *Cissus Tuberosa* ethanol extract (CTE) extract to demonstrate the alkaloids, resins, anthraquinone glycosides, reducing sugars, fats, steroidal glycosides, saponins, tannins was performed by formerly established procedure [23].

### Total phenolic content (TPC)

For the estimation of TPC 0.5 ml of mixture (5ml of distilled water, 0.05g of plant extract), Folin-Ciocalteu reagent (0.5ml) and 7.5ml of distilled water was poured in falcon tube. Placed it at room temperature for ten minutes. After that 20% of $Na_2CO_3$ (1.5ml) was added, incubate for 40 minutes, absorbance was noted at 725 nm. For standard curve, Gallic acid was used TPC were measured as mg/kg of plant extract [24].

## Total flavonoidal content (TFC)

Total flavonoidal content were estimated using quercetin as reference standard. For this procedure 0.5 ml of mixture containing 5ml of distilled water, 0.05g of plant extract kept at 25˚C for some time, After that, 0.6 m of 10% $AlCl_3$ was poured into tube, and left at 27˚C for 15 minutes. Than 2 m of 1 M NaOH was poured in above solution, added 2.4 m of DW in order to produce resulting volume. At 510 nm absorbance was recorded. TFC quantified as quercetin as equivalent [25].

## HPLC analysis

The material has been filtrated using forty-five m PVDF filters, specifically Millex-HV PVDF filters, produced by Millipore in New Bedford, Massachusetts, USA, a thin membrane-like structure. The analysis of the samples was assessed using a Shimadzu chromatographer connected through a tripartite compressor (Shimadzu LC -20AT) and an electron tube sensor using a sampling section (phenomenex® ODS 100 A 250mm, 4.60mm, 5m) and a C18 buffered segment. The processing of data was done using the LC mixture program (version 1.25). In a gradient chromatography processes, both water and acetonitrile were employed as the mobile phase, in the primery acetonitrile/water proportion having (2:8) by volume and the finishing acetonitrile/water proportion having (8:0) ratio in volume, over the course about half hour with a column optimum temperature of 25 degree-celsius and a total volume of injection of 20 pl. The mobile phase was ready for use every day and sonically cleaned to remove gas. 450 to 200 nm wavelengths were used to observe the Ultraviolet spectra. Ethanol was considered to create each and every standard solution and extract. The solution chemical was used at an amount of 2,000 g/ml, while three reference standard solutions were used at various concentrations of 12.5, 25.0 10, 15, 25 50 g/ml [26].

## Experimental animals

Healthy (*n = 30*, male) Wistar albino rats of weight ranging in 100–150 g were considered. All animals observing in study were collected from Government College University's animal house, Faisalabad, Pakistan. All animals facing in separate cages proper nutrition as well as personalized care. All animals were facing sophistically controlled environmental parameters (temperature 25 ˚C ±2˚C, 12-hour light/dark cycles and relative humidity level of 20% to 40%).

## Ethical approval

The guidelines were followed according to National Research Council's, 1996. Before starting the animal study proper approval was taken from committee that deals with animal ethics known as Government College University Faisalabad ethical review committee (ERC), having authorized number GCUF/ ERC/ 266.

## Experimental induction of PD

For the purpose of investigating PD induction, Paraquat (1 mg/kg) was administered intraperitoneally (i.p.) once every five days for a total of 21 days to all animals other than the control group. Following the induction of PQ, the selected appropriate treatment was given after 30 minutes gap [27].

## Study design

All animals were randomly chosen into 6 groups each contain 6 rats

**Group 1:** Labeled as control group (CG) treated with vehicle only

**Group 2:** Labeled as disease control (DC) administered with paraquat (1mg/kg, i.p.), after every 5 days

**Group 3:** Labeled as standard treated (ST) administered with paraquat (1mg/kg, i.p.), after every 5 days + standard drug (levodopa 100mg/kg and carbidopa 25 mg/kg p.o.).

**Group 4:** Administered with CTE 150 mg/kg p.o. + paraquat (1mg/kg, i.p.), after every 5 days

**Group 5:** Administered with CTE 300 mg/kg p.o. + paraquat (1mg/kg, i.p.), after every 5 days

**Group 6:** Administered with CTE 600 mg/kg p.o. + paraquat (1mg/kg, i.p.), after every 5 days

The study trial was conducted for twenty one days. All animals were weighted properly before and after study period, all behavioral parameters was measured properly. All were humanly sacrificed under slight anesthesia by cervical dislocation to minimize pain. Samples were collected for separating serum. All brains were placed in 10% formalin solution and in phosphate buffer for further analysis.

## Behavioral studies

**Open field test.** The test performed by using a wooden square box, measuring 100 cm by 45 cm with a resin-coated floor divided into 25 equal squares by black and red lines. One side of the box was transparent to observe rat movement. Rats were given two minutes to freely explore the box, and researchers recorded their square-sniffing and examination behaviors, including both horizontal and vertical exploration [28].

**Wire hanging test.** The procedure was conducted to analyze neuromuscular coordination, grip strength neuro-muscular stretches of the rat's fore-arms. Experimental description mentioned by Tillerson and Miller 2003 [29]. Rats were trained to grip the wire before the test. Latency time, measuring the duration it took for rats to fall from the wire to the ground surface of the apparatus, was recorded for 120 seconds. This procedure was repeated three times in succession.

**Hole board test.** By watching the rats' head-dipping actions during the hole board test, this experiment attempted to evaluate emotional damage, anxiety, and neophilia in the rats. The setup consisted of a Plexiglas board (20 cm x 20 cm) with 40 cm high walls and 16 equally spaced holes at a height of 1.5 meters above the ground. Rats were allowed eight minutes of exploration on the apparatus's floor. The distance covered by the rats in the center and along the walls was measured and counted head dips, defined as both eyes open, during this time [30].

**Narrow beam walk test.** The study assessed motor coordination using a setup with two small square platforms connected by a 3 cm-wide, 100 cm long beam. Rats started at one end and could move to either side. The number of foot stumbles and the time taken to traverse the beam was recoded [28].

**Y- maze test.** To evaluate memory, cognitive abilities, and spatial cognition in rats, a Y-shaped testing device with three arms (35 x 25 x 10 cm) arranged at 120-degree angles was employed. Rats started at one arm and their unrestricted movement between arms was observed for five minutes. Researchers recorded the total entries into each arm to assess spontaneous behavioral changes [31].

**Estimation of biochemical parameters.** Biochemical parameters were assessed by preparation of tissue homogenate suggested by Saleem et al. [32]. Assessment of malonaldehyde (MDA) level was done by procedure described by Sundas Hira et al. [33]. Evaluation of

superoxide dismutase (SOD) activity was evaluated by Saleem et al. [34]. Catalase (CAT) activity was performed by using previously reported protocol [34] and glutathione (GSH) level was measured by Saleem et al. [35].

## Assessment of neurotransmitters

**Preparation of aqueous phase.**   A 5 ml solution of HCl-n butanol and 50 mg of brain tissues were homogenized together. After centrifuging the mixture at 2000 rpm for ten minutes in order to separate the top layer, 2.50 mlheptanes and 0.30 ml hydrochloric acid (0.1 M) were added while the mixture was constantly agitated. After removing the organic layer, the aqueous layer was separated and employed for further research [36].

**Estimation of noradrenaline and dopamine level.**   To initiate the experiment, a falcon tube was charged with 0.2 mlof an aqueous phase, 0.05 ml of HCl (0.4 M), and 0.1 ml of EDTA. To induce oxidation, introduce 0.1 ml of iodine solution. After a two-minute interval, the reaction was stoped with the addition of 0.1 ml of $Na_2SO_3$ solution. Following another two-minute wait, 0.1 ml of acetic acid was incorporated, and the mixture was then subjected to heating at 100˚C for five minutes. Subsequently, the excitation and emission spectra of the spectrophotometer were recorded after the solution had cooled to room temperature. Dopamine and adrenaline levels were assessed within the wavelength ranges of 330–375 nm and 395–485 nm, respectively [36].

**Estimation of serotonin level.**   In a falcon's test tube, 0.25 ml of the O-phthaldialdehyde (OPT) reagent and 0.25 ml of the aqueous extract were combined. Fluorophore was produced after heating at 100 ˚C for 10 minutes. When the sample reached equilibrium, spectrophotometer values between 350 and 480 nm were obtained. The test solution was concentrated HCl [37].

**Assessment of AChE activity.**   A 15 ml falcon tube was filled with DTNB, 20 L of acetylthiocholine iodide, and 2.6 ml of phosphate buffer at pH 8.0 was introduced into the reaction mixture, followed by the addition of a 0.4 ml sample of tissue homogenate. The absorbance was measured at 412 nm as soon as the initial yellow tint appeared [38].

**Estimation of protein level.**   A 15 ml tube was loaded with brain homogenate (BH), along with 2.5 ml of 2% $Na_2CO_3$ in 0.1 N sodium hydroxide, 1 ml of 1% sodium potassium tartrate in distilled water, and 1 ml of 0.5% copper sulphate solution. This mixture was then allowed to stand for a duration of 10 minutes. The reaction was subsequently allowed to settle for 30 minutes at 37 ˚C after adding 0.5 ml ($H_2O$ and (2N) Folin-Phenol in a 1:2 ratio). At 660 nm, the absorbance was observed, and the protein content was calculated using BSA as the reference [39, 40].

**Evaluation of the histopathological changes in rat brain.**   Brains were kept in 10% formaldehyde solution after removal. Brain tissues fixed in paraffin were sliced into 5 μm thick transverse slices for histopathology. Hematoxylin and eosin dyes were employed for staining the brain sections, and the sample was examined using a 10X light microscope.

## Assessment of inflammatory mediators

**ELISA testing.**   The elabscience ELISA kits for measuring IL-6 (Catalog No: E-ELR0015) and TNF-α (Catalog No: E-ELR0019) were utilised, and all instructions according to manufacturer were followed.

**qRT-PCR assessment.**   This method was carried out for exploring the assertion of α-synuclein, IL-1β, TNF- α, AChE. Table 1 TRIzol technique was performed for RNA extraction. The amount of RNA was calculated by using NanoDrop spectrophotometer and taking absorption at 260–280 nm [41]. cDNA was prepared from RNA using Thermo Scientific cDNA Kit. By

**Table 1. List of primers used in qRT PCR analysis of paraquat-induced Parkinson's, disease rat model.**

| Forward/reverse | Biomarkers | Sequence | Accession No. |
|---|---|---|---|
| Forward | IL-1β | GACTTCACCATGGAACCCGT | |
| Reverse | IL-1β | GGAGACTGCCCATTCTCGAC | NM_031512.2 |
| Forward | AChE | TAGCACCCACTCCATTCICA | |
| Reverse | AChE | TAGCACCCACTCCATTCICA | NM_172009.1 |
| Forward | TNF-α | CTCCCTCCTTGGCCTTTGAA | |
| Reverse | TNF-α | GGAGGGAGAACAGCAACTCC | NM_012675.3 |
| Forward | α- synuclein | TCTGCCAGTTCCACATCTOG | |
| Reverse | α- synuclein | CTCCCTCCTTGGCCTTTGAA | NM_017592500.1 |
| Forward | GADPH | CTCCCTCCTTGGCCTTTGAA | NM_017592435.1 |
| Reverse | GADPH | GCCCATAACCCCCACAACAC | |

using qRT-PCR (Bio-Red system) genes were quantify and amplify. By using Livak method fold changes in biomarkers was evaluated [34].

## Statistical analysis

The findings were shown as Mean ±SD. Software called Graph Pad Prism was used to do the statistical analysis. Tukey's multiple compartment test, and one-way as well as two-way Analysis of Variance (-ANOVA) were considered to determine the minor alterations between the groups. The threshold for statistical significance was ($P < 0.001$).

## Results

### Percentage yield of CTE

By using maceration technique, percentage yield of ethanolic extract of *C.tuberosa* was 2.4%. Total dry extract was found 12g. It was looking like greenish, aromatic semisolid material.

**Phytochemical analysis of CTE.** Proteins, carbohydrates, and lipids were present as main primary phytochemicals. The secondary metabolites, phenolics, tannins, saponins, terpinoides, alkaloids, glycosides, and resins were discovered in phytochemical analysis shown in Table 2.

**Table 2. Result of phytochemical screening of CTE.**

| Constituents | Test performed | Inferences |
|---|---|---|
| Reducing sugar | Fehling's Test | ++ |
| Protein | Biuret test | - |
| Fats | Soap formation test | + |
| Saponins | Frothing test | ++ |
| Steroidal glycosides | Modified Borntrager's test | +++ |
| Tannins | Ferric chloride test | ++ |
| Anthraquinone glycosides | Borntrager's test | +++ |
| Resins | Acid anhydride test | + |
| Alkaloids | Dragendroff's test | +++ |
| Phenolics | Ferric chloride test | +++ |

(+++ = strongly present), (++ = moderately present), (+ = weakly present), (- = not present)

**Table 3. Total phenolic and flavonoids content of CTE.**

| Total Phenolic content (Percentage) | Total Flavonoids content (Percentage) |
|---|---|
| **7.38** | 2.44 |

## Evaluation of phenolic and flavonoid content

In *C. tuberosa*, extract the phenolic, were 7.38%, was more than that of flavonoid compounds, which was found at 2.44%. The entire amount of phenolic compounds was taken by using the suitable formula Y = 0.0716X -0.0135 with R2 = 0.9915 derived from the reference curve of Gallic acid, and the total flavonoids was estimated with regression formula Y = 0.0021X + 0.0149 with R2 = 0.9857 derived by reference quercetin. Result shown in Table 3.

**HPLC analysis.** Five chemicals were positively identified by HPLC analysis of CTE, their structures with name presented in Fig 1 and Table 4. Chlorogenic acid, p-coumaric acid, benzoic acid, gallic acid, and sinapic acid was phytochemicals that were visible with moderate peaks. Fig 1 displays the *C. tuberosa's* HPLC chromatogram.

## Investigation of antiparkinson's activity

**Open field test.** Results revealed that elapsed time in the center and overall number of lines crossed remarkably lower in disease control group than the CTE-treated (P<0.001) shown in Table 5. Treatment with CTE significantly and dose-dependently increased the exploratory behaviour of rats.

**Wire hanging test.** Findings show that the disease control group's neuromuscular strength was remarkably lower (P<0.001) than control. Results demonstrated that high doses of extract, significantly boosted rates' neuromuscular strength. This improvement in hanging time was indicated in the Fig 2(a).

**Hole board test.** The findings showed that the disease control group had considerably fewer head dipping cases than both the standard group and control (P<0.001). Cognitive capacities as well as memory functioning of the CTE treatment groups significantly

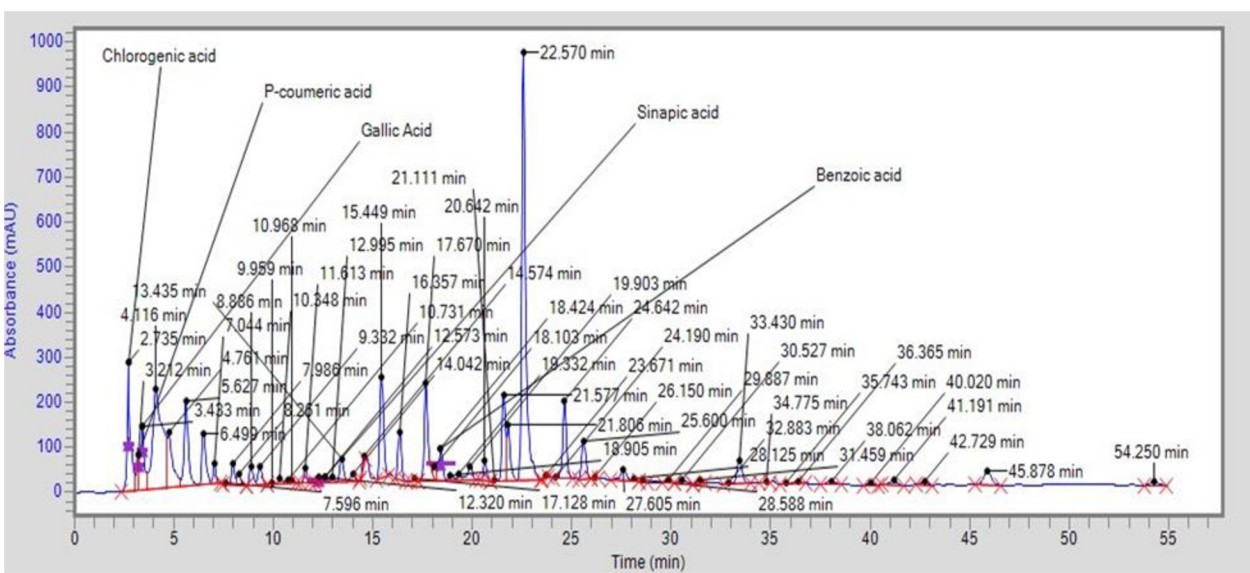

**Fig 1. HPLC chromatogram of CTE.**

**Table 4. HPLC screening of CTE.**

| Sr. No. | Compounds | Retention Time (min) | Amount in ppm | Pharmacological activity | Reference |
|---|---|---|---|---|---|
| 1 | Chlorogenic acid | 2.735 | 3.55 | Anti Neuro-inflammatory | [66–68] |
| | | | | Neuro-protective | |
| | | | | Anti-Parkinson's effect | |
| | | | | Bost up mental health | |
| | | | | Reduce Alzheimer's disease | |
| 2 | P-coumaric acid | 3.212 | 4.11 | Anti-Alzheimer's | [69, 70] |
| | | | | Anti-Parkinson | |
| | | | | Improve cognitive abilities, | |
| | | | | Neuroprotective | |
| 3 | Gallic Acid | 3.433 | 3.02 | Reduce neurodegeneration | [71–73] |
| | | | | Anti-neuroinflammatory | |
| | | | | Support brain health | |
| | | | | Reduce the symptoms of dementia | |
| 4 | Sinapic acid | 12.573 | 5.12 | Anti-neuroinflammatory | [74–76] |
| | | | | Enhance brain functioning | |
| | | | | Anti-Alzheimer's | |
| | | | | Remove neurodegeneration | |
| 5 | Benzoic acid | 18.424 | 2.35 | Neuroprotective agent | [77–79] |
| | | | | Improve memory functioning | |
| | | | | Anti-Alzheimer's | |
| | | | | Anti-Parkinson | |

($P<0.001$) improve as increased dose levels (600 > 300 > 150 mg/kg). The findings were shown in Fig 2(b).

**Y-maze test.** The study's findings demonstrate that the disease control group's percent alteration and number of arm entries were considerably less ($P<0.001$) than control. The percentage change and the overall number of arm entries both demonstrate significant ($P<0.001$) outcomes for CTE administered levels. By the provided dose level, the results demonstrated the greatest memory recovery at a treatment level of 600 mg/kg. Fig 3 include results.

**Narrow beam walk test.** The findings of this test, which measured motor coordination and stability, are shown in Fig 4. The disease control exhibited higher latency time to travel the distance from one end of beam to another end compared to control group ($P <0.001$). The standard group has dramatic variance, with time latency time that significantly shorter

**Table 5. Effect of CTE on open field test.**

| Groups | Number of lines crossed | Time spent in central area |
|---|---|---|
| **Control** | 16.633±1.706 | 54.167±2.358 |
| **Disease Control** | 5.000±1.065[###] | 9.667±2.404[###] |
| **Standard treated** | 24.333±1.687[***] | 52.500±2.964[***] |
| **CTE 150mg/kg** | 13.167±1.167[**] | 25.167±3.219[**] |
| **CTE 300mg/kg** | 15.833±1.249[***] | 35.167±2.701[***] |
| **CTE 600mg/kg** | 23.667±1.626[***] | 49.500±2.604[***] |

Statistically significant data reported as the mean value ± SEM, sample interval (n) = 6, [###]($P < .001$) in comparison to control group, and [***]($P<0.001$) compared to disease control, ns indicated non-significant.

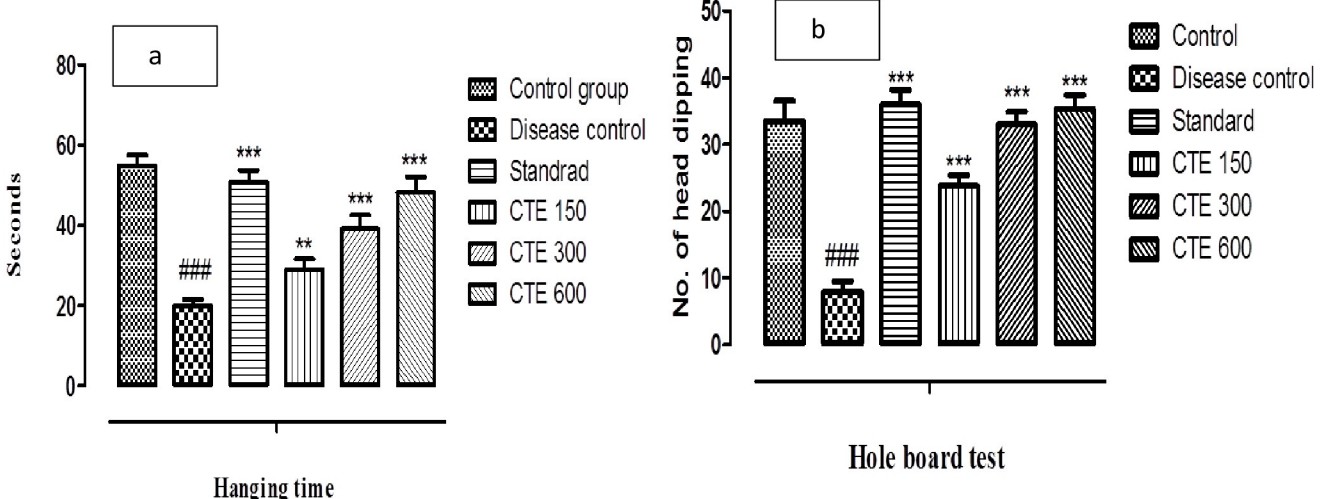

**Fig 2. (a)** Effect of CTE on wire hanging experiment. **(b)** Effect of CTE on hole board experiment. Statistically significant values reported as the mean ± SEM, sample size (n) = 6, $^{###}$(P <0 .001) with comparison to control group, and $^{***}$(P<0.001) disease control, ns indicated non-significant.

(P <0.001). In treatment groups, there was a dose-dependent reduction in time latency to cross the beam (P < 0.001) (Fig 4).

**Evaluation of antioxidant enzymes in brain homogenate.** As shown in Table 6, the disease control group's lipid peroxidation (MDA level) was substantially higher than control. But in disease control, all of the measures (SOD, CAT, and GSH level) were considerably (P<0.001) lowered compared to control, standard and CTE treatment groups. SOD, CAT, and GSH levels were enhanced in the CTE and standard treated groups, but MDA levels significantly decreased (P < .001) compared to disease control group.

**Evaluation of neurotransmitter levels.** In Table 7 results of dopamine, serotonin and noradrenaline of all considered groups are mentioned. Results indicated that remarkable reduction counted in dopamine, serotonin and noradrenaline levels in diseased group than normal control. CTE treated groups displayed dose variation responses (dopamine level increased by increasing dose).

Fig 5(A) represented, AChE levels were substantially higher in disease control than control (P < .001). When compared to disease control, the AChE level was considerably (P < .001) decreased in CTE-treated groups.

Fig 5(B) reported protein level was considerably lower (P < .001) in diseased groups. CTE administered groups have a dose variation response, with 600 mg/kg demonstrating higher levels of normal protein than minimum dosage level. CTE treated groups displayed significantly raised (P < .001) level of protein in rats brain.

**Histopathological analysis of brain.** According to the histological examination of the brain, the paraquat-treated (disease control) group's brain tissue architecture was abnormal compared to that of the control group. Significant neuronal injury, astrogliosis, astrocytes hypertrophy, nuclear pyknosis, neural loss, Lewy body formation, hemorrhage, inflammation, pigmentation, wide intracellular distance, and programmed cell death are detected in diseased group. Neuronal damage was significantly reduced in the group that received conventional treatment (L-dopa, carbidopa). In contrast to 300 and 150 mg/kg, the 600 mg/kg CTE-treated animals exhibit an amazing reversal of neuronal damage in dose variation manner (Fig 6).

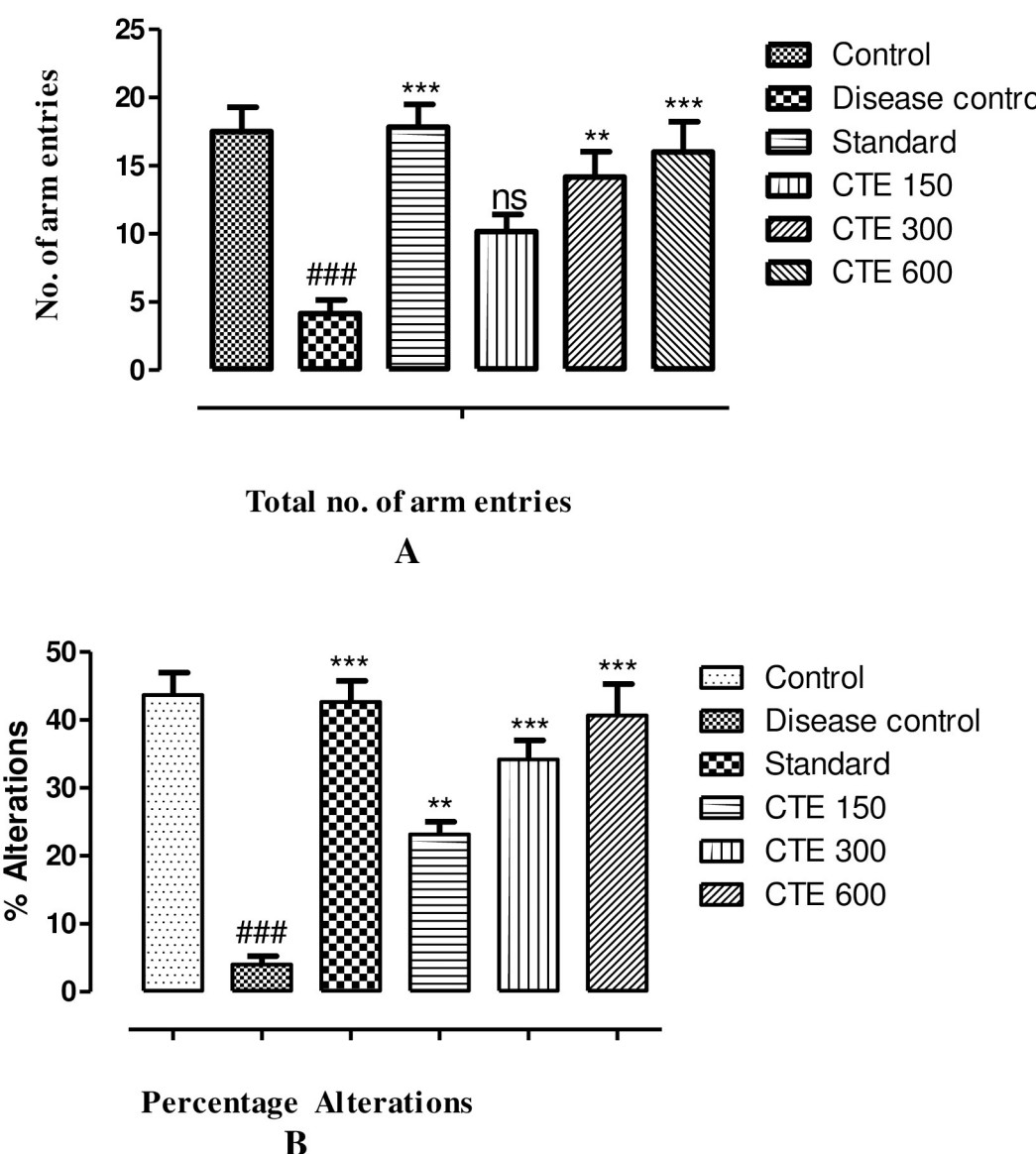

**Fig 3. Result of Y-maze test (A) Number of arm entries (B) alterations.** Statistically significant values reported as the mean ± SEM, sample size (n) = 6, ###(P <0 .001) with comparison to control group, and ***(P<0.001) disease control, ns indicated non-significant.

**Assessment of inflammatory biomarkers by ELISA.** The results indicated a noteworthy enhancement (P > .001) in the IL-6 and TNF-α levels in the disease control group when compared to the control group, as presented in Table 8. The effects observed in the CTE-treated groups demonstrated a progressive alteration depending on the dosage. In contrast to the disease control group, the treatment group receiving 600 mg/kg exhibited a considerable decrease (P > .001) in the IL-6 and TNF-α levels.

## Gene expression analysis by qRT-PCR

In the paraquat-induced rat model, the mRNA expression of various PD biomarkers was notably increased. TNF-α, α-synuclein, AChE, and IL-1β showed significant restoration (P>0.001)

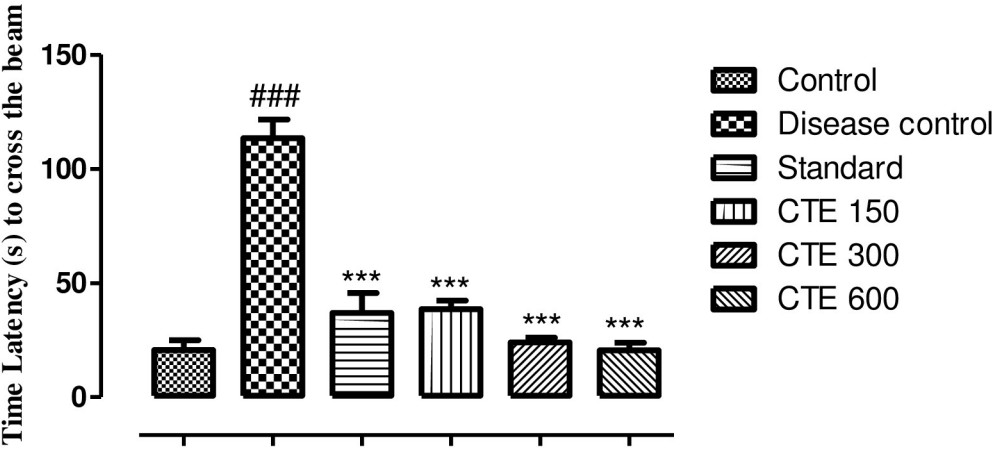

**Narrow beem walk test**

**Fig 4. Effect of CTE on narrow beam walk test.** Statistically significant values reported as the mean ± SEM, sample size (n) = 6, $^{###}$(P < .001) with comparison to control group, and $^{***}$(P<0.001) disease control, ns indicated non-significant.

**Table 6. Effect of CTE on first line antioxidant enzymes in rat model of paraquat intoxication.**

| Groups | CAT (IU/µl) | SOD (IU/µl) | MDA(TBA mg/kg) | GSH(µg/mg protein) |
|---|---|---|---|---|
| Control | 0.518±0.022 | 1.385±0.129 | 2.913±0.025 | 0.472±0.049 |
| Disease Control | 0.046±0.013$^{###}$ | 0.593±0.022$^{###}$ | 4.652±0.023$^{###}$ | 0.173±0.021$^{###}$ |
| Standard treated | 0.403±0.029$^{***}$ | 0.923±0.023$^{ns}$ | 3.465±0.027$^{***}$ | 0.497±0.032$^{***}$ |
| CTE 150mg/kg | 0.323±0.021$^{***}$ | 0.668±0.022$^{ns}$ | 2.912±0.022$^{***}$ | 0.350±0.015$^{**}$ |
| CTE 300mg/kg | 0.402±0.024$^{***}$ | 0.735±0.018$^{ns}$ | 3.320±0.050$^{***}$ | 0.435±0.017$^{***}$ |
| CTE 600mg/kg | 0.447±0.018$^{***}$ | 0.834±0.146$^{ns}$ | 3.795±0.063$^{***}$ | 0.468±0.021$^{***}$ |

Statistically significant data reported as the mean value ± SEM, sample interval (n) = 6, $^{###}$(P < .001) with comparison to control group, and $^{***}$(P<0.001) disease control, ns indicated non-significant.

**Table 7. Effect of CTE on neurotransmitter level in rat model of paraquat intoxication.**

| Groups | Dopammine (µg/µg of bt) | Noradrenaline (µg/µg of bt) | Serotonin (µg/µg of bt) |
|---|---|---|---|
| Control | 0.341±0.004 | 0.054±0.004 | 0.626±0.004 |
| Disease Control | 0.059±0.031$^{###}$ | 0.026±0.003$^{###}$ | 0.352±0.009$^{###}$ |
| Standard treated | 0.531±0.013$^{***}$ | 0.073±0.003$^{***}$ | 0.585±0.004$^{***}$ |
| CTE 150mg/kg | 0.324±0.006$^{***}$ | 0.036±0.002$^{ns}$ | 0.327±0.005$^{*}$ |
| CTE 300mg/kg | 0.423±0.007$^{***}$ | 0.037±0.003$^{ns}$ | 0.369±0.005$^{ns}$ |
| CTE 600mg/kg | 0.512±0.005$^{***}$ | 0.054±0.003$^{***}$ | 0.535±0.004$^{***}$ |

Statistically significant data reported as the mean value ± SEM, sample interval (n) = 6, $^{###}$(P < .001) with comparison to control group, and $^{***}$(P<0.001) disease control, ns indicated non-significant.

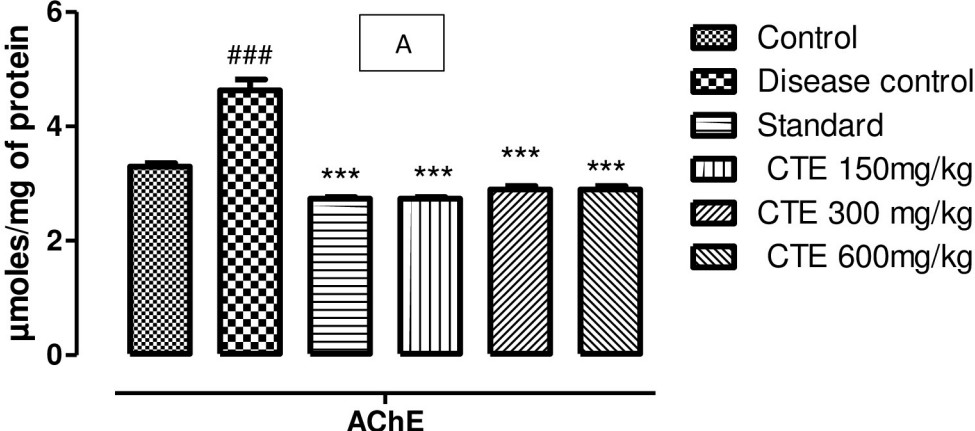

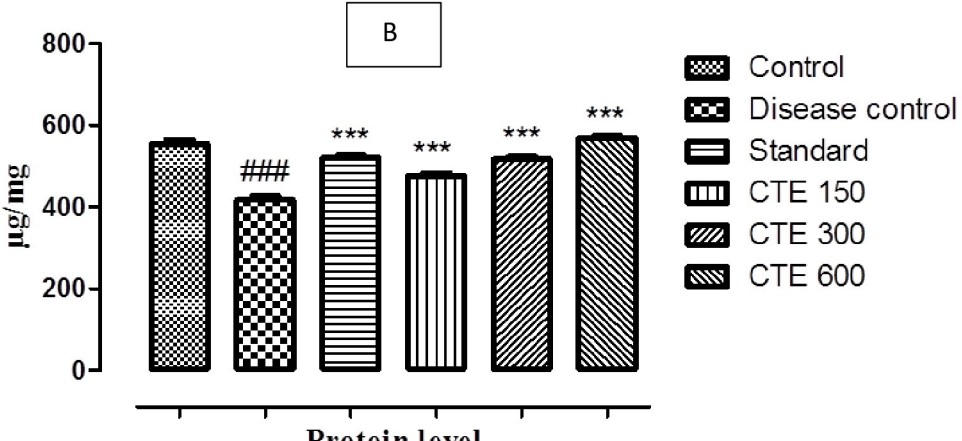

**Fig 5. (A)** Effect of CTE on of acetylcholinesterase level, Fig 6(B) Effect of CTE on of protein level. Statistically significant values reported as the mean ± SEM, sample size (n) = 6, ###(P < .001) with comparison to normal control group, and ***(P<0.001) disease control, ns indicated non-significant.

in the treatment groups receiving 600 mg/kg, as well as in those administered 150 mg/kg and 300 mg/kg, and in the standard treatment group (L-dopa + carbidopa) compared to disease control group (Fig 7).

## Discussion

The prevalence of neurodegenerative disorders is increasing at an alarming rate, with more and more cases being reported continuously. Age dependence is a significant factor in the majority of neurodegenerative illnesses, with a close relationship between age and disease prevalence [42]. There is a dearth of specialized treatments, increased morbidity, and expensive medications for neurodegenerative illness; hence research of novel herbal neuro-protective substances is crucial.

The confirmation of secondary metabolites by phytochemical investigation of *C. tuberosa*, revealed the plant's medicinal significance. Phenolic-rich dietary supplements and plant

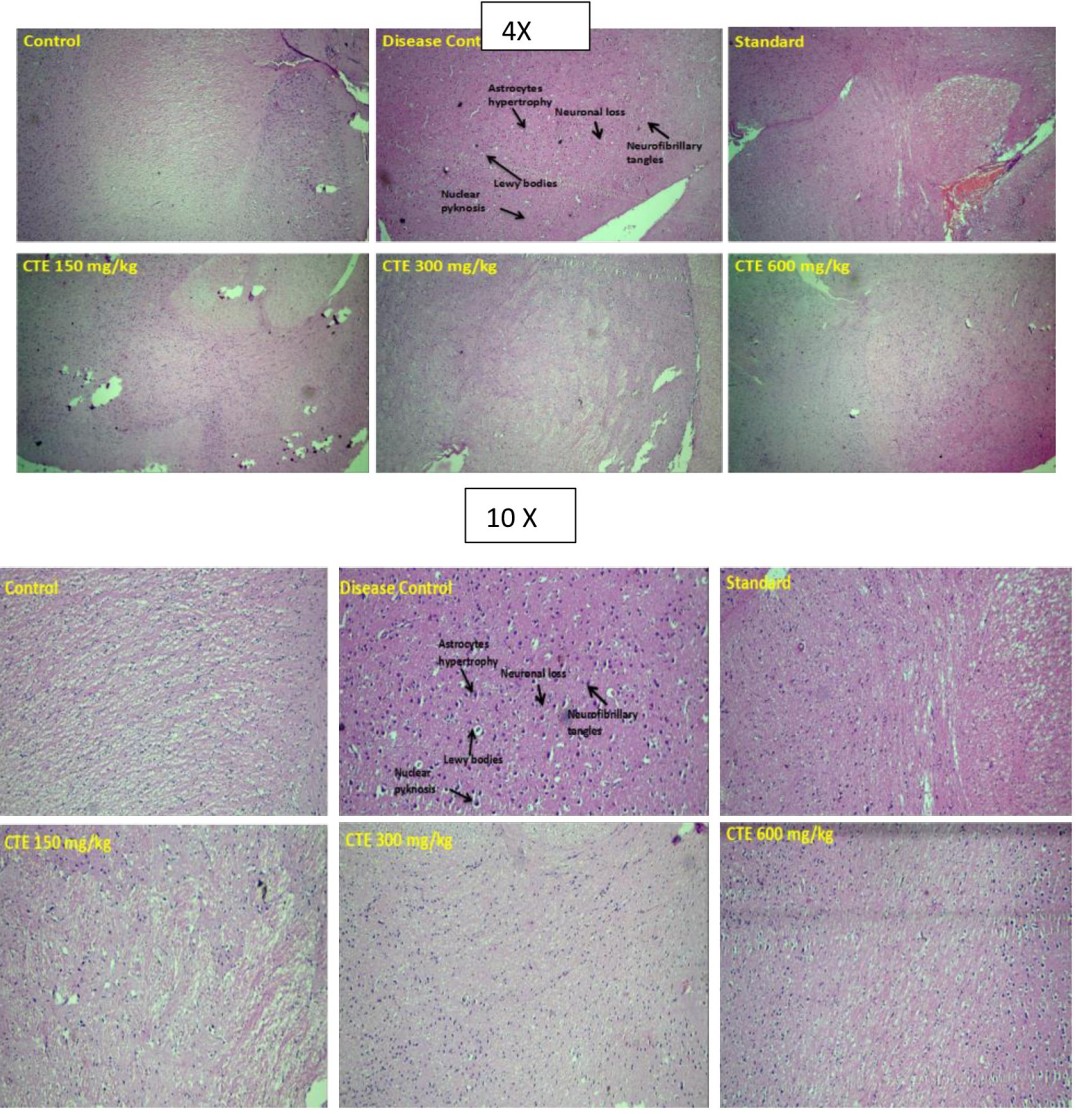

**Fig 6. Brain histopathological analysis at (4X and 10X) of substantia nigra subjects treated with paraquat.** Notably, the levels of LB, NFT, IP and AP were significantly reduced in a group that received standard treatment. The impact of neurodegeneration was moderately mitigated in those treated with a 150 mg/kg CTE. A marked improvement in neuronal loss and PG was observed in the 300 mg/kg CTE-treated group. The 600 mg/kg CTE-treated group displayed remarkable enhancements in rectifying the structural abnormalities within the brain sections.

**Table 8. Effect of inflammatory biomarkers (TNF-α and IL- 6) in rat model of paraquat intoxication.**

| Groups | TNF-α level (pg/ml) | IL-6 level (pg/ml) |
|---|---|---|
| Control | 1046.000±4.719 | 421.489±2.583 |
| Disease Control | 3876.667±5.931### | 542.810±2.949### |
| Standard treated | 1863.333±22.755*** | 150.837±2.994*** |
| CTE 150mg/kg | 3036.667±8.751*** | 461.696±3.676*** |
| CTE 300mg/kg | 2050.000±5.893*** | 299.845±3.329*** |
| CTE 600mg/kg | 1032.333±11.857*** | 167.906±2.361*** |

Statistically significant data reported as the mean value ± SEM, sample interval (n) = 6, ###($P < .001$) with comparison to control group, and ***($P<0.001$) compared to disease control, ns indicated non-significant.

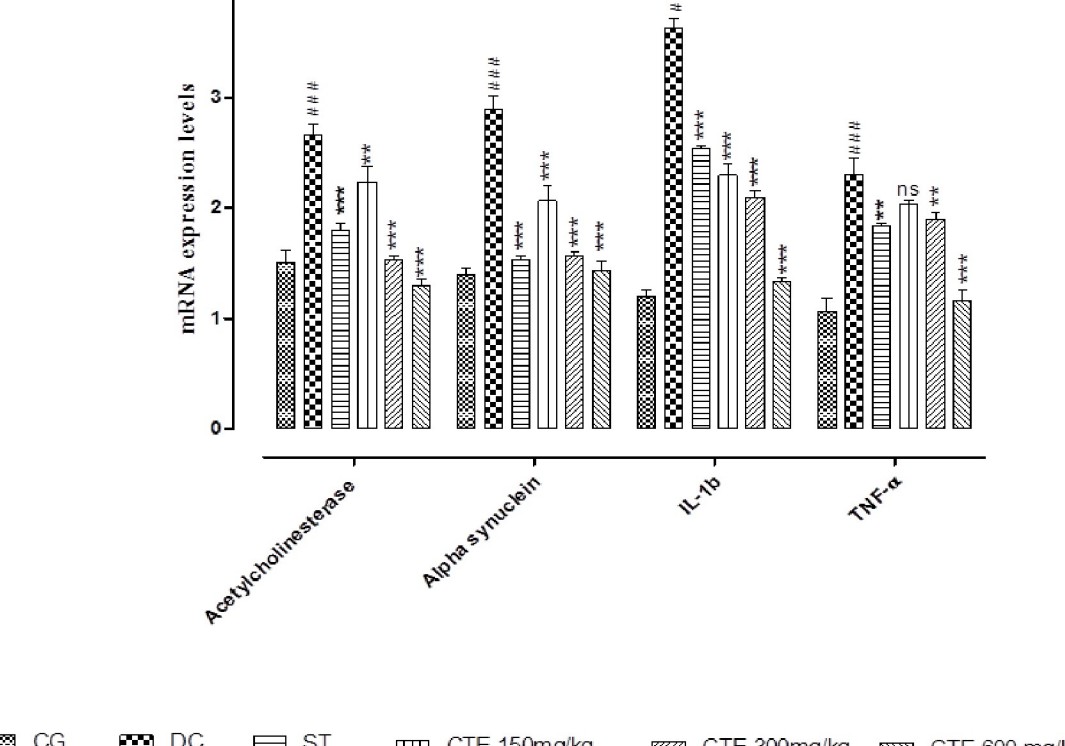

**Fig 7. Effect of CTE on mRNA expression of miscellaneous pathological biomarkers in PD rat.** Where CG; Control group, DC; Disease control group; ST; Standard treated, 150-, 300-, and 600 mg/kg are CTE treatment groups, IL-1b; interleukin 1 beta, TNF-α; tumor necrosis factor alpha. Results are presented as mean ± SD (n = 06) and, $^{###}$(P < 0.001) with comparison to control group, and $^{***}$(P<0.001) compared to disease control, ns indicated non-significant.

extracts have the capacity to mitigate neuro-inflammation by modulating intracellular signaling pathways that enhance cell survival and age-related neuronal function [43]. The phenolic found in *C. tuberosa* are the main ingredients for the development of neuroprotective agents because they promote neuronal regeneration and reduce oxidative stress. Findings of HPLC analysis showed proof of the antioxidant and neuroprotective properties of active ingredients found in *C. tuberosa*. Among antioxidants, the most superior ones include phenolic acids like chlorogenic acid and p-coumaric acid, exhibiting neuroprotective effect through attenuation of oxidative insult in brain tissues.

In the disease control group, paraquat induced neuro-inflammation in the SN zone by entering the central nervous system through the neutral protein transporter. The results of the wire hanging and hole board tests demonstrate that CTE (at doses of 150, 300, and 600 mg/kg) enhances hanging time and head dipping in the treatment group, showcasing a dose-dependent improvement in neuromuscular coordination and a reduction in depression and psychological difficulties associated with PD. Thus, the wire hanging test outcomes align with a previous study conducted by Jay Parkash and Satyndra Kumar Yadav, wherein the extract from *Withania somnifera* root significantly prolonged hanging time and mitigated neuroinflammation in the paraquat-induced Parkinson's disease model [44]. The hole board and Y maze tests were performed as elaborated by Bagewadi et al., and Ogunnoiki et al., CTE treatment significantly improved the exploratory, locomotory and spatial memory in treated animals [45, 46].

Our findings are also consistent to a study which investigated the effects of curcumin, niacin, and a new drug (ZM241385) on PD in mice. The study found that combining these treatments effectively improved behavioral and biochemical symptoms, with ZM241385 showing promising results similar to the established drug Sinemet®. Additionally, curcumin and niacin supplements showed efficacy in alleviating PD symptoms [47].

Disruptions in the regulation of redox potential within neurons hinder multiple biological processes and, in the end, result in cellular demise [48]. The body safeguards itself from the impacts of free radicals by employing an antioxidant defense system, which includes SOD, CAT, and GSH. The redox-cycling compound paraquat elevates the levels of Reactive oxygen species (ROS), leading to mitochondrial damage and, ultimately, oxidative stress. As a result of impaired mitochondrial function, the group exposed to paraquat exhibited an imbalance in lipid peroxidation level [49]. The MDA levels rise in the disease-control groups, while marked decline in lipid peroxidation was manifested in CTE-treated groups. The levels of the antioxidant enzymes plays a critical role in determining the endogenous antioxidant capacity and cellular homeostasis [50]. In the present study, there is a substantial reduction ($p < 0.001$) in the levels of these antioxidant enzymes when compared to the disease control group. Conversely, CTE treatment significantly boosts these indicators of antioxidants [51]. As per the study findings, the administration of a high dose of CTE (600 mg/kg) led to a substantial increase in protein levels when compared to the disease control group. Our results are congruent to previous study investigated the effectiveness of various treatments for rotenone-induced PD in rats, including natural sources (hesperidin and naringenin) and pharmaceutical agents (levodopa and ZM241385). Naringenin showed the most promising results, with ZM241385 and levodopa exhibiting similar levels of improvement [52]. Similarly our results are parallel to study in which *Citrus sinensis* peels extract was evaluated as a potential anti-Parkinsonism agent in rats, showing promising results in restoring neurotransmitter levels, antioxidant parameters, and other biomarkers. The extract was rich in bioactive compounds, including phenolic acids, flavonoids, and polymethoxylated flavones, and exhibited a more potent effect than L-dopa. This study suggests the potential development of new nutraceuticals and pharmaceutical agents, as well as functional food products, for the treatment of PD [53].

Dopamine, noradrenaline, serotonin, and acetylcholine are essential neuromodulators within the basal ganglia, responsible for regulating motor and cognitive functions. Neuro-inflammation leads to a decline in the levels of these neurotransmitters. Additionally, the degeneration of choline acetyltransferase results in reduced acetylcholine levels, further impairing cognitive function in PD. This degeneration contributes to the loss of dopaminergic neurons in the pars compacta. AChE inhibitors play a pivotal role in preventing neurodegeneration by mitigating cell death [54]. Owing to the presence of phenolic compounds, the rat group treated with CTE exhibited a noteworthy increase ($p < 0.001$) in the levels of these neurotransmitters. In particular, the significant CTE dose (600 mg/kg) revealed a substantial elevation in acetylcholine levels, achieved by inhibiting the enzyme acetylcholinesterase responsible for breaking down acetylcholine into acetic acid and choline. One of the findings from our study is consistent with a prior investigation in which Ashwagandha root juice demonstrated its anti-Parkinsonian effects by reducing oxidative stress through a decrease in lipid peroxidation in a Paraquat-induced model [55]. In light of these findings, it was concluded that CTE enhanced behavioral parameters as seen by enhanced memory, motility, and physical strength.

The pathogenic dysfunctions of PD are significantly influenced by the presence of a mutation in the SNCA gene, leading to the excessive expression of the α-synuclein protein. This protein is being explored as a potential early diagnostic biomarker and therapeutic target for PD in current scientific research approaches. Previous studies have indicated that cinnamon extract can effectively inhibit the aggregation of α-synuclein and stabilize oligomers in both *in*

*vitro* and *in vivo* settings, particularly in models of PD induced by α-synuclein. The findings of Siddique et al. align with our own research, demonstrating that *Eucalyptus citriodora* improved climbing abilities and reduced oxidative stress in a drosophila model of PD [56]. Consistent with the aforementioned studies, CTE (presumably another treatment) improved the mRNA expression of α-synuclein in the treatment groups when compared to the model group. This improvement is attributed to its various secondary metabolites.

CTE similar to *Mucuna pruriens* exhibited the protective effect against paraquat induced neurodegeneration in Wistar rats. *M. pruriens* are traditionally used in Asian countries in treatment of PD. The presence of favorable amount of L-Dopa (L-dihydroxyphenylalanine) making them a robust natural remedy for PD hallmarks. Manyam et al. reported the neuroprotective effect of *M. pruriens* extract at dose 16 mg/kg in unilateral 6-hydroxydopamine lesion by mitigating the sensory and motor deficit. *M. pruriens* extract at higher dose 48mg/kg induced contralateral turning behavior, it also antagonized tacrine induced Parkinsonian tremors [57]. CTE treatment improved motor and non-motor symptoms in parkinsonian rats due to presence of *M. pruriens* like phytochemicals in extract. In a study *M. pruriens* extract showed the *in vitro* acetylcholinestersae inhibitory potential in comparison to L-dopa which increased the activity of this enzyme, suggesting superior protective potential in neurodegenerative disorders compared to L-Dopa [58, 59]. CTE showed the ant-neuroinflammatory potential through modulation of neuroinflammatory biomarkers. CTE markedly downregualted the mRNA expression of IL-1α, IL-1β, TNF-α and α- synuclein suggesting therapeutic role in neurodegeneration like *M. pruriens.. pruriens* significantly decreased the neuroinflammation induced by -methyl-4-phenyl-1,2,3,6-tetrahydropyridine (MPTP) intoxication in mice through modulation of Glial Fibrillary Acidic Protein, TNF-α and inducible nitric oxide synthase in substania nigra. *M. pruriens* inhibited the NF-κB signaling and activate pAkt1 activity and retarded the dopaminergic neuronal apoptosis. *M. pruriens* also decreased the oxidative stress by increasing the level of first line antioxidant enzymes in nigrostriatl region. HPLC analysis revealed the presence of neuroactive compounds like chlorogenic acid, P-coumaric acid, gallic Acid, benzoic acid and sinapic acid. Presence of these compounds in sufficient amount made CTE a marked therapeutic entity against neuroinflammation, impairment in behavioral dysfunctions, cognitive failure and in cataleptic conditions induced by neurotoxins. The HPLC analysis revealed the presence of neuro-protective phytochemicals in extract of *M. pruriens* which palliate MPTP induced neurotoxicity in mice [60]. *M. pruriens* seed extract showed neuroprotective effect in multiple toxins induced PD models due to presence of sufficient amount of L-Dopa [61]. It was reported that *Withania somnifera* treatment mitigated the behavioral impairment induced by BPA in mice and also recovered the NMDA receptor linked to memory and cognition in hippocampus region [62] Acetylcholinesterase (AChE) primarily functions to terminate cholinergic neurotransmission at synapses by breaking down acetylcholine. Previous research consistently indicates that an increased expression of a specific synaptic isoform of AChE mRNA is associated with the initiation of apoptosis in various cell lines. In the PD model group, the neurotoxic effects of paraquat resulted in elevated levels of AChE and its mRNA expression. In contrast, the treatment group administered with CTE exhibited a significant reduction in AChE expression compared to the model group. These findings suggest that CTE has therapeutic potential in preventing apoptotic cell death in dopaminergic neurons, aligning with prior research results.

As evidenced in previous research, the development of PD is strongly linked to an increased expression of pro-inflammatory cytokines such as TNF-α and IL-1β. This increase is triggered by the activation of microglial cells in the striatum and cerebrospinal fluid (CSF). It has been reported that the sustained elevation of the pro-inflammatory cytokine IL-1β is responsible for the death of dopaminergic neurons in the PD brain. Furthermore, elevated levels of TNF-α

have been associated with forelimb akinesia and neurodegeneration. CTE, in the treatment groups, effectively mitigated the heightened levels of proinflammatory cytokines TNF-α and IL-1β in comparison to the disease control group. This effect is attributed to CTE's ability to inhibit neuroinflammation and reduce oxidative stress, aligning with previous research findings.

The prior literature demonstrated that elevated expression of pro-inflammatory cytokines like TNF-α and IL-6 is closely associated with PD development. An increase in TNF-α has the potential to increase the levels of certain cytokines that may contribute to neurodegeneration [63]. TNF-α and IL-6, as pro-inflammatory cytokines, contribute to the demise of dopaminergic cells in the PD-afflicted brain. These cytokines serve as potent microglial activators that are involved in the death of dopaminergic neurons [64]. The pro-inflammatory cytokines TNF-α and IL-6 promote the loss of dopaminergic cells in the PD-affected brain. These cytokines act as robust microglial activators, leading to the decline of dopaminergic neurons [65]. In the present study, the disease control group exhibited elevated levels of TNF-α and IL-6 due to dysfunction in the brain's basal ganglia region [47]. CTE displayed dose-dependent anti-inflammatory effects, with the most significant impact observed at a dose of 600 mg/kg (p < 0.001).

Microglia cells, serving as the brain's intrinsic macrophages, become activated when exposed to neurotoxic substances such as paraquat. In PD, these cells experience widespread activation in the SN, leading to the release of inflammatory cytokines and the initiation of neuro-inflammation [52]. In this context, TNF-α and IL-1β hold key roles as mediators of microglial functions, and in the pathogenesis of PD, they are chiefly responsible for the demise of dopaminergic neurons [53]. The increased mRNA expression of TNF-α at the site of neuronal damage suggests its potential as a therapeutic target for addressing this neurological disorder. The current study reveals that CTE, particularly at a high dose, significantly reduces the mRNA expression of TNF-α.

Recent advancements in research have unveiled the potential to manage neurodegenerative disorders and slow down their progression using anti-inflammatory agents, including medicinal plants that can inhibit microglial activation. The influence of CTE on the control of inflammatory cytokines aligns with the effects observed in a methanol extract derived from *F. religiosa* leaves, which demonstrated a dose-dependent reduction in the mRNA expression of TNF-α, IL-1β, and IL-6 in mice. IL-1β plays a role in regulating neuroinflammation by impacting the up-regulation of MAPK pathways while simultaneously suppressing NF-b levels.

The findings of this study have important implications, specifically that CTE has shown potential neuroprotective effects in a paraquat-induced PD model through multifaceted mechanistic pathways such as by mitigation of neuroinflammation and oxidative stress. CTE boosted behavioral parameters, including memory, motility, and physical strength. Neuro-chemical analysis has revealed that CTE raised the levels of neurotransmitters, such as dopamine, noradrenaline, serotonin, and acetylcholine. CTE retarded the aggregation of α-synuclein and stabilized oligomers with decreased expression of pro-inflammatory cytokines, such as TNF-α and IL-1β in dose dependent manner. These findings suggest that CTE may be a promising candidate for the development of novel neuroprotective agents for the treatment of PD. The study's results are consistent with previous research on the potential therapeutic effects of herbal extracts in neurodegenerative disorders.

## Conclusion

The conclusion of recent systematic investigation from qualitative, quantitative analysis, biochemical, behavioral, histopathological alterations, and toxicity profiling of *C. tuberosa*

maintained its remarkable potential to deal paraquat induced neuroinflammation in PD. It elevated the levels of primary antioxidant enzymes while decreasing AChE, α-synuclein, IL-6, IL-1β, and TNF-α. Therefore, *C. tuberosa* should be regarded as a promising candidate for novel drug development, potentially serving as a foundation for the formulation of neuroprotective agents.

## Author Contributions

**Conceptualization:** Malik Saadullah, Amna Sehar, Rida Siddique, Aisha Sethi.

**Data curation:** Malik Saadullah, Rida Siddique, Muhammad Asif, Shazia Anwer Bukhari, Aisha Sethi.

**Formal analysis:** Rida Siddique, Hafsa Tariq, Shazia Anwer Bukhari, Aisha Sethi.

**Funding acquisition:** Malik Saadullah.

**Investigation:** Malik Saadullah.

**Methodology:** Malik Saadullah, Amna Sehar, Zunera Chauhdary.

**Project administration:** Malik Saadullah.

**Resources:** Malik Saadullah, Shazia Anwer Bukhari, Aisha Sethi.

**Software:** Rida Siddique, Shazia Anwer Bukhari, Aisha Sethi.

**Supervision:** Rida Siddique, Shazia Anwer Bukhari, Aisha Sethi.

**Validation:** Rida Siddique, Shazia Anwer Bukhari, Aisha Sethi.

**Visualization:** Rida Siddique, Shazia Anwer Bukhari, Aisha Sethi.

**Writing – original draft:** Amna Sehar, Rida Siddique.

**Writing – review & editing:** Amna Sehar, Rida Siddique, Hafsa Tariq, Shazia Anwer Bukhari.

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
