## [Decision Letter · Decision Letter 0]

1 May 2024

PONE-D-24-00220Anti-neuroinflammatory and neuroprotective potential of Cissus Tuberosa ethanol extract in Parkinson’s disease model through the modulation of neuroinflammatory markersPLOS ONE

Dear Dr. Chauhdary,

Thank you for submitting your manuscript to PLOS ONE. After careful consideration, we feel that it has merit but does not fully meet PLOS ONE’s publication criteria as it currently stands. Therefore, we invite you to submit a revised version of the manuscript that addresses the points raised during the review process. Please submit your revised manuscript by Jun 15 2024 11:59PM. If you will need more time than this to complete your revisions, please reply to this message or contact the journal office at plosone@plos.org. Please include the following items when submitting your revised manuscript:A rebuttal letter that responds to each point raised by the academic editor and reviewer(s). You should upload this letter as a separate file labeled 'Response to Reviewers'.A marked-up copy of your manuscript that highlights changes made to the original version. You should upload this as a separate file labeled 'Revised Manuscript with Track Changes'.An unmarked version of your revised paper without tracked changes. You should upload this as a separate file labeled 'Manuscript'.

We look forward to receiving your revised manuscript.

Kind regards,

Phakkharawat Sittiprapaporn, Ph.D.

Academic Editor

PLOS ONE

Journal Requirements:

2. We note that your Data Availability Statement is currently as follows: 

"All relevant data are within the manuscript and its Supporting Information files."

5. Please ensure that you refer to Figures 5, 7, and 8 in your text as, if accepted, production will need this reference to link the reader to the figure.

6. We note you have included a table to which you do not refer in the text of your manuscript. Please ensure that you refer to Table 1 in your text; if accepted, production will need this reference to link the reader to the Table.

**Additional Editor Comments:**

Although your paper is of interest, the reviewers suggest major revisions and have not recommended publication of the manuscript in its present form, especially with some missing references and citations.  We encourage you to consider these comments and make an appropriate revision of your manuscript. The reviewers' comments are below. 

Reviewers' comments:

Reviewer's Responses to Questions

**Comments to the Author**

1. Is the manuscript technically sound, and do the data support the conclusions?

Reviewer #1: Yes

Reviewer #2: Yes

Reviewer #3: Yes

2. Has the statistical analysis been performed appropriately and rigorously? 

Reviewer #1: Yes

Reviewer #2: Yes

Reviewer #3: Yes

3. Have the authors made all data underlying the findings in their manuscript fully available?

Reviewer #1: Yes

Reviewer #2: Yes

Reviewer #3: Yes

4. Is the manuscript presented in an intelligible fashion and written in standard English?

Reviewer #1: Yes

Reviewer #2: Yes

Reviewer #3: Yes

5. Review Comments to the Author

Reviewer #1: Dear author

In figure 5, please use white arrows and write the abbreviation in white color for more clear observation. Also write the name of each slide (liver, heart,….) in white color as the black color not appear due to the dark tissue architectures.

In figure 7, please identify the brain region of these histopathology observations.

I encourage you to cite these references in the discussion section.

1- Motawi TK, Sadik NAH, Hamed MA, Ali SA, Khalil WKB, Ahmed YR. 2020. Potential therapeutic effects of antagonizing adenosine A2A receptor, curcumin and niacin in rotenone-induced Parkinson's disease mice model. Mol. Cell. Biochem., 465:89–102.

2- Hamed MA, Aboul Naser AF, Aziz WM, Ibrahim FM, Ali SA, El-Rigal NS, Khalil WKB. 2020. Natural sources, dopaminergic and non-dopaminergic agents for therapeutic assessment of Parkinsonism in rat model. PharmaNutrition, 11 (2020), 100171.

3- Hamed MA, Aboul Naser AF, Elbatanony MM, El-Feky AM, Matloub AA, El-Rigal NS, Khalil WKB. 2021. Therapeutic potential of Citrus sinensis peels against rotenone induced Parkinsonism in rats. Curr. Bioactive Comp., 17 (6), e: 010621186105.

4- Ahmed YR, Aboul Naser AF, Elbatanony MM, El-Feky AM, khalil WKB, Hamed MA. 2023. Gene expression, oxidative stress, and neurotransmitters in rotenone-induced Parkinson’s disease in rats: Role of naringin from Citrus aurantium via blocking adenosine A2A receptor. Curr. Bioactive Comp., e101023221984. DOI: 10.2174/0115734072268296231002060839.

Reviewer #2: Dear Dr.,

Title: Anti-neuroinflammatory and neuroprotective potential of Cissus tuberosa ethanol extract in Parkinson’s disease model through the modulation of neuroinflammatory markers

Manuscript ID: PONE-D-24-00220

Overall comments: Saadullah et al., described in this manuscript: the role of ethanolic extract of Cissus tuberosa in the Parkinson’s disease model via modulation of neuroinflammatory reactions. The limitation of this research work is recent references are missing. The overall manuscript is good and it can help those working in this field of research.

Specific comments:

1. The abstract is written well. Keywords need to be mentioned with the most appropriate words.

2. Abbreviations were appearing for the first time; need to mention the full form.

3. The introduction section has too many small paragraphs. Need to write sequential and logical manner.

4. Material and Methods section: subsection 2.3.2 must appear before 2.3.1; similarly 2.3.3. can merge with 2.3.2.

5. Oral toxicity of CTE can be placed before the PD induction study.

6. The order of behavioual study needs to describe; why selected. What is the interference effect between behavioural assessments?

7. 6 groups and in each group, 6 rats were used for the PD study. How to maintain the statistical data n=6 maintained for biochemical assessment and even author assessed histology evaluation made.

8. How statistical significance P < 0.001 was achieved for all assessments.

9. Figure 5 legend must mention the scale bar and magnification used.

10. The effect of CTE 600mg/kg was shown more than Standard treatment in CAT (IU/μl)

11. The spelling error ‘Dopammine’ needs to be corrected throughout the manuscript.

12. Discussion can be made with concise statements.

13. Limitations and future scope of the study can be incorporated.

14. References are too old references; need to be updated with recent and relevant references.

Minor comments

1. The too many small paragraphs can be merged and typo errors need to be rectified.

2. References are too old; and need to be updated.

*****

Reviewer #3: Dear Author,

The current manuscript that was about to anti-neuroinflammatory and neuroprotective potential of Cissus Tuberosa ethanol

extract in Parkinson’s disease model is very well written and I did not find any special writing or scientific problems in it.

6. PLOS authors have the option to publish the peer review history of their article (what does this mean?). If published, this will include your full peer review and any attached files.

Reviewer #1: **Yes: **Manal A. Hamed, Prof. of Biochemistry, National Research Centre, Cairo, Egypt.

Reviewer #2: No

Reviewer #3: No

---

## [Author Response · Author response to Decision Letter 0]

20 May 2024

Thank you so musch for valueable suggestions. i revise manuscript as suggested, kindly check and proceeed, i will be highly thankful to you

Dear editor we use ethical statement only in method section in main manuscript, we did not use it at another section. kindly consider our revision now, i will be highly thankful, as we need it

---

## [Decision Letter · Decision Letter 1]

20 Aug 2024

PONE-D-24-00220R1Anti-neuroinflammatory and neuroprotective potential of Cissus Tuberosa ethanol extract in Parkinson’s disease model through the modulation of neuroinflammatory markersPLOS ONE

Dear Dr. Chauhdary,

Thank you for submitting your manuscript to PLOS ONE. Your manuscript, referenced above, has now been reviewed by experts in the field.  After careful consideration, we feel that it has merit but does not fully meet PLOS ONE’s publication criteria as it currently stands. The comments of the reviewers are included below in order for you to understand the basis for our decision, and we hope that their thoughtful comments will help you in your future studies. Specific issues raised by our experts in the field include “inconsistent use of abbreviations and full terms; discrepancies in the representation of numbers and units; results section presented in a list-like manner, lacking coherence and clear interpretation; and discussion section that lacks focus and does not clearly highlight the significance of the findings, etc. Given these significant issues, this manuscript might not be appropriate for publication in its current form. We recommend that extensive editing for language, writing structure, and consistency be necessary before it can be considered for publication. Therefore, we invite you to submit a revised version of the manuscript that addresses the points raised during the review process.

If applicable, we recommend that you deposit your laboratory protocols in protocols.io to enhance the reproducibility of your results. Protocols.io assigns your protocol its own identifier (DOI) so that it can be cited independently in the future. For instructions, see: https://journals.plos.org/plosone/s/submission-guidelines#loc-laboratory-protocols. Additionally, PLOS ONE offers an option for publishing peer-reviewed Lab Protocol articles, which describe protocols hosted on protocols.io. Read more information on sharing protocols at https://plos.org/protocols?utm_medium=editorial-email&utm_source=authorletters&utm_campaign=protocols.

We look forward to receiving your revised manuscript.

Kind regards,

Phakkharawat Sittiprapaporn, Ph.D.

Academic Editor

PLOS ONE

Reviewers' comments:

Reviewer's Responses to Questions

**Comments to the Author**

1. If the authors have adequately addressed your comments raised in a previous round of review and you feel that this manuscript is now acceptable for publication, you may indicate that here to bypass the “Comments to the Author” section, enter your conflict of interest statement in the “Confidential to Editor” section, and submit your "Accept" recommendation.

Reviewer #4: (No Response)

2. Is the manuscript technically sound, and do the data support the conclusions?

Reviewer #4: Partly

3. Has the statistical analysis been performed appropriately and rigorously? 

Reviewer #4: No

4. Have the authors made all data underlying the findings in their manuscript fully available?

Reviewer #4: Yes

5. Is the manuscript presented in an intelligible fashion and written in standard English?

Reviewer #4: No

6. Review Comments to the Author

Reviewer #4: The study investigated the neuroprotective properties and safety of the ethanolic extract of Cissus tuberosa (CTE), which is rich in phenolic compounds. The research aimed to confirm the anti-Parkinson’s activity of CTE by examining its effects on α-synuclein, interleukin-1β (IL-1β), and tumor necrosis factor-α (TNF-α) levels. Parkinson’s disease was induced in test subjects using paraquat, and various experimental groups were established for behavioral and biochemical analysis.

The results demonstrated that Cissus tuberosa extract significantly improved paraquat-induced neurotoxicity without showing signs of toxicity in histopathological analysis. Additionally, the biochemical markers and neurotransmitter levels were improved, and the expression of inflammation-related genes was reduced. These findings suggest that Cissus tuberosa has potential in alleviating symptoms of Parkinson’s disease.

Despite the promising findings, the manuscript contains significant issues in its English language and overall writing style. The numerous grammatical errors, unclear phrasing, and inconsistent formatting impede the readability and comprehension of the study. Specific issues include:

1. Inconsistent use of abbreviations and full terms.

2. Discrepancies in the representation of numbers and units.

3. Tense inconsistencies throughout the manuscript.

4. Results section presented in a list-like manner, lacking coherence and clear interpretation.

5. A Discussion section that lacks focus and does not clearly highlight the significance of the findings.

6. Use of “%age” instead of “percentage.”

7. Unclear distinctions between ### (P < .001) and *** (P < 0.001) in Table 8.

8. Inconsistent use of “AchE” and “AChE.”

9. Lack of clear differences explained between Control, Disease control, and standard in Figures 2, 3, and 4.

10. Incorrect terminology, such as “p-Coumeric acid” instead of “p-Coumaric acid.”

7. PLOS authors have the option to publish the peer review history of their article (what does this mean?). If published, this will include your full peer review and any attached files.

Reviewer #4: No

---

## [Author Response · Author response to Decision Letter 1]

24 Aug 2024

Dear editor i made all suggested changes in manuscript, i also attached track change file. Dear editor please publish my article as soon as possible. i need this article for my annual performance. thanks waiting anxiously for your positive response

---

## [Editor Report · Decision Letter 2]

4 Sep 2024

PONE-D-24-00220R2Anti-neuroinflammatory and neuroprotective potential of Cissus Tuberosa ethanol extract in Parkinson’s disease model through the modulation of neuroinflammatory markersPLOS ONE

Dear Dr. Chauhdary,

Thank you for submitting your manuscript to PLOS ONE. After careful consideration, we feel that it has merit but does not fully meet PLOS ONE’s publication criteria as it currently stands. Therefore, we invite you to submit a revised version of the manuscript that addresses the points raised during the review process.

We look forward to receiving your revised manuscript.

Kind regards,

Sachchida Nand Rai, Ph.D.

Academic Editor

PLOS ONE

Journal Requirements:

Additional Editor Comments:

The study's objectives and outcomes lack clear alignment, making it difficult to discern whether the authors have met their goals. The inclusion of toxicity studies across various organs appears unnecessary and detracts from the focus of the research. Moreover, the histopathological data presented are unconvincing, and the HPLC data lacks robustness and clarity. The authors' responses to the reviewers' comments are vague and do not adequately address the concerns raised.

A significant omission in the manuscript is the absence of lower magnification images of the brain's histopathological data. Specifically, the substantia nigra and striatum, which are crucial for assessing Parkinsonian pathology, should be clearly visualized at 4x and 10x magnifications. The authors must provide histopathological images at these magnifications to ensure the data's reliability and accuracy.

Furthermore, In addition, authors should correlate the Anti-Parkinsonian activity of Cissus Tuberosa ethanol extract with Mucuna pruriens. Mucuna pruriens exhibits strong Anti-Parkinsonian activity in MPTP intoxicated mouse model along with Anti-apopototic activity of Withania somnifera in Praquat and Maneb intoxicated mouse model. The discussion section also lacks coherence and fails to adequately reference relevant, previously published studies. A thorough revision of the manuscript is necessary to address these issues, ensuring that the study's outcomes are clearly aligned with its objectives and supported by convincing data.

---

## [Author Response · Author response to Decision Letter 2]

10 Sep 2024

Dear editor i make new images as suggested at 4X and 10 X magnification, remove grammatical mistakes as suggested add citation and improve discussion. kindly proceed my article i need to publish this article in this year , necessary for my evaluation at work place please. i will be highly thankful to you

---

## [Editor Report · Decision Letter 3]

16 Sep 2024

Anti-neuroinflammatory and neuroprotective potential of Cissus tuberosa ethanol extract in Parkinson’s disease model through the modulation of neuroinflammatory markers

PONE-D-24-00220R3

Dear Dr. Zunera,

The manuscript has been meticulously revised in accordance with both the reviewers' feedback and my own suggestions. All necessary modifications have been implemented to improve the clarity, quality, and overall presentation of the research. Given the comprehensive nature of these revisions, I believe the manuscript now meets the required standards for publication. 

Kind regards,

Sachchida Nand Rai, Ph.D.

Academic Editor

PLOS ONE

Additional Editor Comments (optional):

The manuscript has been meticulously revised in accordance with both the reviewers' feedback and my own suggestions. All necessary modifications have been implemented to improve the clarity, quality, and overall presentation of the research. Given the comprehensive nature of these revisions, I believe the manuscript now meets the required standards for publication.
---

## [Editor Report · Acceptance letter]

23 Sep 2024

PONE-D-24-00220R3 

PLOS ONE

Dear Dr. Chauhdary, 

I'm pleased to inform you that your manuscript has been deemed suitable for publication in PLOS ONE. Congratulations! Your manuscript is now being handed over to our production team.

Kind regards, 

on behalf of

Dr. Sachchida Nand Rai 

Academic Editor

PLOS ONE